# Identification of Viruses in *Molossus* Bats from the Brazilian Amazon: A Descriptive Metagenomic Analysis

**DOI:** 10.3390/microorganisms12030593

**Published:** 2024-03-16

**Authors:** Lucas Rafael Santana Pinheiro, Érika Dayane Leal Rodrigues, Francisco Amilton dos Santos Paiva, Ana Cecília Ribeiro Cruz, Daniele Barbosa de Almeida Medeiros, Alexandre do Rosário Casseb, Sandro Patroca da Silva, Livia Medeiros Neves Casseb

**Affiliations:** 1Graduate Program in Virology, Hemorrhagic Fevers and Arbovirology Section (SAARB), Evandro Chagas Institute (IEC), BR-316 Highway, km 7, Levilândia, Ananindeua 67030-000, PA, Brazil; lucaspinheiro523@hotmail.com; 2Hemorrhagic Fevers and Arbovirology Section (SAARB), Evandro Chagas Institute (IEC), BR-316 Highway, km 7, Levilândia, Ananindeua 67030-000, PA, Brazil; erika_dayane.lr@hotmail.com (É.D.L.R.); franciscopaiva@iec.gov.br (F.A.d.S.P.); anacecilia@iec.gov.br (A.C.R.C.); dbamedeiros@gmail.com (D.B.d.A.M.); spatroca@gmail.com (S.P.d.S.); 3Health and Animal Production Institute, Federal and Rural University of Amazon (UFRA), Belém 66077-830, PA, Brazil; alexcasseb@yahoo.com.br

**Keywords:** metagenomics, Amazon, bat, *Paramyxoviridae*, *Dicistroviridae*, morbillivirus

## Abstract

Bats are widely distributed in Brazil, including the Amazon region, and their association with viral pathogens is well-known. This work aimed to evaluate the metavirome in samples of *Molossus* sp. bats captured in the Brazilian Amazon from 2019 to 2021. Lung samples from 58 bats were divided into 13 pools for RNA isolation and sequencing followed by bioinformatic analysis. The *Retroviridae* family showed the highest abundance of viral reads. Although no complete genome could be recovered, the *Paramyxoviridae* and *Dicistroviridae* families showed the formation of contigs with satisfactory identity and size characteristics for further analysis. One contig of the *Paramyxoviridae* family was characterized as belonging to the genus *Morbillivirus*, being grouped most closely phylogenetically to *Porcine morbillivirus*. The contig related to the *Dicistroviridae* family was identified within the *Cripavirus* genus, with 94%, 91%, and 42% amino acid identity with *Culex dicistrovirus* 2, *Rhopalosiphum padi*, and *Aphid lethal paralysis*, respectively. The presence of viruses in bats needs constant updating since the study was able to identify viral sequences related to families or genera still poorly described in the literature in association with bats.

## 1. Introduction

Bats of the *Molossus* genus (*Molossidae* family) are insectivorous animals found all over the American continent. In Brazil, records of their occurrence have been described in all regions, including the Amazon region. Specimens of the genus are found in both urban and rural areas [1,2,3]. Viral investigations in *Molossus* spp. carried out in Brazil and other South American countries provide evidence of infection by viruses classified into several families, such as *Nairoviridae*, *Picornaviridae*, *Astroviridae*, *Retroviridae*, *Coronaviridae*, *Rhabdoviridae*, and *Paramyxoviridae* [4,5,6].

*Paramyxoviridae* is a family of enveloped, negative-sense, and single-stranded RNA viruses. The representatives of this family have genomes ranging from 14,296 to 20,148 nucleotide (nt). The subfamily *Orthoparamyxovirinae* is where this genus is included, along with the genera *Aquaparamyxovirus*, *Ferlavirus*, *Henipavirus*, *Jeilongvirus*, *Morbillivirus*, *Narmovirus*, *Respirovirus*, and *Salemvirus* [7], which encompass some species known to be pathogenic, either in humans or in animals, such as *Hendra henipavirus*, *Nipah henipavirus*, *Human respirovirus* 1, *Human respirovirus* 3, *Murine respirovirus*, *Bovine respirovirus* 3, and others [8].

*Morbillivirus* representatives encode eight proteins: phosphoprotein (P), large polymerase protein (L), RNA-binding protein, matrix protein (M), fusion protein (F), and an attachment protein (H) as structural proteins, and C and V as non-structural proteins [9]. They are also well-known agents responsible for causing diseases in their natural hosts. Among them are *Measles morbillivirus* (the causative agent of measles) and *Canine morbillivirus* (the causative agent of canine distemper) [10,11].

A morbillivirus infection can occur through the respiratory tract when the host inhales suspended viral particles. These particles can be present in the air because of the aerosols generated by other contaminated hosts, for example, through urine [12]. The first association between bats and morbilliviruses occurred in research by Drexler et al. [6].

Members of the *Dicistroviridae* family have a positive-sense RNA genome, ranging from 8500 to 10,000 nt in length, comprising two ORFs that decode the non-structural (ORF1) and structural (ORF2) proteins of the viruses [13]. Representatives of this family infect primarily arthropods, such as bees and flies, but they are also present, for example, in shrimp [14]. However, viruses of this family have already been detected in fecal samples from vertebrate animals, such as geese [15]. In bats, these viruses have also been detected, either in feces [16,17], tissue samples [13,18], or even in blood [19].

The purpose of this study was to evaluate the viral diversity among bats of the *Molossus* sp. genus living in the Amazon region, aiming at the identification of viral pathogen presence or new virus species, especially regarding human or veterinary potential health importance.

## 2. Materials and Methods

### 2.1. Samples

Lung samples were obtained from 58 bats of the *Molossus molossus* and *Molossus rufus* species that were captured, from 2019 to 2021, in five Brazilian states: Pará, Rondônia, Amazonas, Roraima, and Tocantins. All bat samples were sent to the Department of Arbovirology and Hemorrhagic Fevers (SAARB) in Evandro Chagas Institute (IEC) by units of the health departments of states and municipalities in accordance with the rabies surveillance program and were submitted to direct immunofluorescence and mouse inoculation tests for the *Rabies virus* with negative results.

This study consisted of a descriptive and univariate metagenomic investigation with non-probabilistic samples selected by the following factors: genus of bat (*Molossus* sp.), states of the Brazilian Amazon region, and years of collection. To visualize the collection points and sample distribution, a map was constructed using the R program v4.3.0 with the help of the ggplot2 and geobr libraries [20], and using the Brazilian Institute of Geography and Statistics (IBGE) 2020 database (Figure 1).

### 2.2. Sample Extraction, Synthesis of cDNA and Genomic Library Preparation

To perform the extraction of RNA, lung samples were taken from the quiropterygians. 13 pools were formed, containing tissues from up to 5 bats per pool. The pools were formed based on the year and state of bats collection.

For the extraction of the viscera, approximately 10 mg of tissue was placed in a 2 mL microtube with 1 mL of TRIzol Plus RNA Purification Kit (Thermo Fischer Scientific Inc., Waltham, MA, USA) and a 5-mm diameter steel ball and then agitated in the TissueLyser II equipment for 2 min at a frequency of 25 Hz. At the end of this process, the samples were taken to a biological safety cabinet to proceed with the extraction of the genetic material.

In the same tube, 200 µL of chloroform were added, and then all this material was transferred to a tube containing a phase separator polymer (Phasemaker™ Tubes-Thermo Fischer Scientific Inc., Waltham, MA, USA) and then the separation procedure were carried out according to manufacturer’s recommendations. At the end, approximately 560 µL of the phase above the polymer were transferred to a new 1.5 mL microtube with the subsequent addition of 530 µL of 70% ethanol. The contents of this new tube were purified using the PureLink RNA Mini Kit (Thermo Fischer Scientific Inc., Waltham, MA, USA), following the manufacturer’s recommendations.

After purification, the RNA was quantified using the Qubit RNA HS Assay Kit (Thermo Fischer Scientific Inc., Waltham, MA, USA) along with the Qubit 4.0 equipment, following the manufacturer’s recommendations.

The preparation of cDNA from RNA started with the first and second strand cDNA synthesis, using the SuperScript^TM^ VILO^TM^ MasterMix kit (Thermo Fischer Scientific Inc., Waltham, MA, USA) and NEBNext^®^ Second Strand Synthesis Module (New England Biolabs Inc., Ipswich, MA, USA) respectively. The cDNA purification was performed by the PureLink^®^ PCR Purification Kit (Thermo Fischer Scientific Inc., Waltham, MA, USA). All steps followed the recommendations of the manufacturer of the respective kits. The synthesized cDNA was quantified using the Assay DNA HS Kit (Thermo Fischer Scientific Inc., Waltham, MA, USA) in the Qubit 4.0 equipment.

The genomic library was prepared according to the guidelines of the Nextera XT DNA kit (Illumina, Inc., San Diego, CA, USA) and sequenced using the NextSeq 500 platform (Illumina, Inc., San Diego, CA, USA) using the NextSeq 500/550 High Output Kit v2.5 (300 cycles) (Illumina, Inc., San Diego, CA, USA), using the paired-end methodology, according to the manufacturer’s recommendations.

### 2.3. Bioinformatics

#### 2.3.1. Filtering, Evaluation, and Processing of Readings

The fastq files were obtained after demultiplexing the sequencing output files (BCL-Basecall file) using the bcl2fastq software v1.8.4. As a result of this process, two fastq files were generated, R1 and R2, which correspond to the raw reads for each sample.

Initially, the quality of the reads originated from the sequencing step was evaluated using the Fastp program v0.23.2 [21], with the removal of short reads (less than 75 nt), adaptor fragments, and reads with indeterminate bases (reads with more than 10% N), Next, the Bowtie2 program v2.4.5 [22] was used to remove host reads using the *Molossus molossus* [23] reference genome mMolMol1.p (Aug. 2020 Genome Browser—GCF_014108415.1 assembly) available in NCBI and access by UCSC Genome Browser Gateway (https://genome.ucsc.edu/cgi-bin/hgGateway, accessed on 15 August 2022). Finally, SortMeRNA v.2.1 [24] was used for ribosomal RNA (rRNA) removal. At the end of the treatment, analysis with the Fastp program was applied comparatively to evaluate the efficiency of the data treatment.

Subsequently, the DIAMOND software v2.0.15 [25] was applied to annotate the reads for each file, using the viruses’ database of non-redundant (nr) proteins, considering an e-value of 1 × 10^−4^ and amino acid identity. The output files generated by DIAMOND were visualized in Krona v2.8 [26]. The results obtained were tabulated and saved in a file with the extension csv, and these tables were loaded and evaluated in the R program v4.3.0 through the heatmaply library in order to observe patterns of abundance of the different viruses found for each of the sequenced samples. The Kraken program v2.1.2 was also employed, using the nr database and displaying results in the Pavian program v1.0.

#### 2.3.2. De Novo Assembly, Prediction and Gene Annotation

The files generated in the treatment step were used for assembly through De Novo methodology using SPAdes v3.13.1 [27] and MEGAHIT v1.2.9 [28] programs, with k-mer values of 21, 33, 55, and 77; and of 21, 31, 41, 51, 61, 71, 81, 91, and 99, respectively. The clustered contigs were analyzed using DIAMOND v2.0.15 [25] for BlastX analysis utilizing the viruses nr protein database, considering an e-value of 1 × 10^−4^ and amino acid identity values.

After a more general analysis in Krona v2.8.1 to filter possible viruses of interest, the files relative to the contigs generated for each pool were visualized in the Megan program v6.24.11, using an e-value filter of 1 × 10^−10^.

The contigs that presented similarity with viral sequences underwent gene prediction analysis using the GeneMarkS tool [29]. The amino acid sequences of the probable genes were compared to different protein databases available in the InterProScan5 program [30], which performs a search for protein domains. The tabulated data were inspected for viral domains of interest and curated by the program Geneious, version 9.1.8 (https://www.geneious.com/, accessed on 29 August 2022).

### 2.4. Phylogenetic Analysis

Phylogenetic inference was performed according to the analysis of the amino acid sequences obtained from the samples and the sequences available in the international database National Center for Biotechnology Information (http://www.ncbi.nlm.nih.gov, accessed on 5 September 2022), using the coding regions of non-structural proteins.

The dataset generated alongside the samples from the study was submitted to Multiple Sequence Alignment (MSA) using Mafft v.7 [31]. The result of the alignment was inspected manually to perform corrections when necessary, using the program Geneious v.9.1.8 (https://www.geneious.com/, accessed on 6 September 2022).

Initially, the aligned dataset was submitted to analysis to identify the best nucleotide substitution model and verify the phylogenetic signal. Then, the phylogenetic trees were constructed using Maximum Likelihood (ML) methodology [32]. These methodologies were employed using the IQ-TREE v.1.6.12 program [33]. In conjunction with these analyses, the bootstrap test was applied, setting 1000 replicates in order to provide greater reliability to the grouping values [34].

The phylogeny visualization was performed by the FigTree v.1.4.4 software. The data set used did not have a root sequence, so the methodology of midpoint rooting was used, a tool available in the phylogeny visualization program.

After assessment and editing of the phylogeny, a file with the extension “sgv” (Scalable Vector Graphics) was generated for editing and manipulation of the image using the Inkscape v.1.1 program (https://inkscape.org/, accessed on 26 September 2022).

## 3. Results

### 3.1. Overall Results

The sequencing of the 13 sample pools resulted in a total of 830,981,690 reads. Of these, 74,805 corresponded to RNA viral sequences. Table 1 shows the quantity of reads generated for each pool and how many of them remained after the data processing steps.

The treated reads have their identity distribution displayed in the Appendix A. Most reads were identified as string sequences (12.15%), followed by bacteria (7.91%), fungi (1.72%), protozoa (0.38%), and, to a lesser extent, viruses, which accounted for only 0.072% of the reads treated. More than 70% of the reads could not be identified.

Among the viral reads, 32 families were identified, distributed among the 13 pools. The families *Poxviridae*, *Mimiviridae*, *Retroviridae*, *Siphoviridae*, and *Myoviridae* were the only ones to have compatible reads in all pools. Among them, *Retroviridae* was the most represented, followed by *Siphoviridae* and *Myoviridae*, in that order. The *Herpesviridae*, *Picornaviridae*, *Phycodnaviridae*, *Inoviridae*, and *Podoviridae* families were also widely detected.

Pools 7 (5RO20) and 12 (4TO19) showed the highest diversity in viral families (18 in total) and were also the only pools with a positive detection of reads for the *Dicistroviridae* family. Pool 4 (5PA21) had the lowest viral diversity, with only eight families identified by reads.

The families Nimaviridae, Rhabdoviridae, Drexlerviridae, Permutotetraviridae, Solemoviridae, Microviridae, Genomoviridae, Tymoviridae, Totiviridae, Iflaviridae, and Paramyxoviridae all had positivity in only one pool. This information is displayed in Figure 2.

Regarding viral diversity per state, Rondônia was the most diverse with twenty-one viral families total, while Pará had twenty, Tocantins eighteen, Roraima fourteen, and Amazonas nine.

A total of 5,605,280 contigs were generated, of which slightly more than 13% were from MEGAHIT and slightly less than 87% from SPADES (Table 2). Of this total, only 0.8% had a positive correlation for viruses, with pool 5 (4PA20/21) presenting the most virus-positive contigs and pool 6 (5RO19) presenting the fewest.

### 3.2. Morbillivirus Identification

The identity analysis implemented via BlastX of a contig measuring 1142 bp relative to pool 3 (5PA20) showed amino acid identity of 85.77%, e-value = 0.0, and query cover of 99% with L protein from *Porcine morbillivirus* (GenBank QWQ56143). After refinement of the analysis by reference map, it was possible to extend the sequence to 1498 bp, equivalent to 499 amino acids. The nucleotide sequence comprised 9.56% of the *Porcine morbillivirus* genome (GenBank MT511667), with 6× coverage, ranging from position 11,810 to 13,380, referring to a portion of the L gene.

In addition to the identity with *Porcine morbillivirus*, it was possible to observe the proximity with other viruses from the identity matrix in the Appendix A, both amino acid and nucleotide, which revealed identity over 70% (nucleotide) and 80% (amino acid) between the sequence obtained in this study with three other viral genomes belonging to the *Morbillivirus* genus, namely *Canine morbillivirus*, *Phyllostomus bat morbillivirus*, and *Phocine morbillivirus*. Allied to this data, the analysis of protein motifs displayed in the Appendix A showed very similar and conserved regions between the studied sequence and the sequences of ten other morbilliviruses. The studied sequence was then named *Molossus bat morbillivirus* (MBMV).

The analysis of the protein domains revealed a strong similarity between the MBMV domain positions and domains of other morbillivirus sequences, indicating that the portion recovered in the present study was a part of the RNA-dependent RNA polymerase (RdRp) domain (Figure 3).

The phylogenetic tree (Figure 4) was generated using reference sequences (refseq) for viruses of all *Paramyxoviridae* family genera, plus two reference sequences of bat morbilliviruses and one sequence of *Porcine morbillivirus*, which was identified as the most similar to the sequence of interest in a curation step using the BlastX tool of NCBI. The dataset assembled to generate the phylogenetic tree presented a phylogenetic signal above 60% in the summation of the vertices and less than 30% in the central triangle.

It is possible to identify that the MBMV sequence forms a monophyletic clade with other viral species belonging to the *Morbillivirus* genus, being more closely related to *Phyllostomus bat morbillivirus* and *Porcine morbillivirus sequences*, although on a more external branch in comparison to these two viruses.

### 3.3. Dicistroviridae

The Dicistroviridae family showed identity, confirmed by BlastX, to a contig from pool 7 (5RO20) containing 1426 nt with 95% amino acid identity to the non-structural polyprotein (GenBank AXQ04777) of *Culex dicistrovirus* 2 (GenBank MH188005; e-value = 0; Query cover = 97%). Attempts to extend the sequence using a reference map were unsuccessful.

An identity matrix was obtained and is presented in the Appendix A, where 91% and 42% amino acid similarity were obtained with *Rhopalosiphum padi* and *Aphid lethal paralysis* viruses, respectively, and 94% identity to *Culex dicistrovirus* 2, which is not a reference sequence for this family. Then, it was possible to assemble a phylogenetic tree using ORF1 related to the non-structural polyprotein of *Dicistroviridae* reference sequences. The sequence obtained presented a helicase protein domain from InterProScan analysis (Figure 5) and formed a monophyletic clade within the *Cripavirus* genus (Figure 6), being more closely related to *Culex dicistrovirus* 2 and *Rhopalosiphum padi virus*. Given these results, the genomic fragment in question was identified as a *Culex dicistrovirus* 2 strain.

The database used to assemble the phylogeny presented a value above 60% in the summation of the vertices and less than 30% in the central triangle from the phylogenetic signal test, and the alignment of the sequences is presented in the Appendix A.

## 4. Discussion

### 4.1. Overall Discussion

*Microviridae*, *Siphoviridae*, *Myoviridae*, *Podoviridae*, *Herelleviridae*, *Drexlerviridae*, *Inoviridae*, *Ackermannviridae*, and *Salasmaviridae* are viral families infecting bacteria. In accordance with the study objectives, bacteriophage sequences were not investigated further. Encountering bacterial virus sequences is common in studies of chiropteran viromes [18,35,36].

Solemoviridae, Tymoviridae, Genomoviridae, Totiviridae, Marseilleviridae, Mitoviridae, Mimiviridae, Phycodnaviridae, and Partitiviridae are viral families related to infections in fungi, protozoa, amoebas, algae, and plants. The presence of these viruses here is unlikely to come from the occurrence of viremia. Another issue reinforcing this idea is that the number of readings generated for these families was low, like family Solemoviridae represented by four reads only. A higher abundance of viral genome would be expected in the event of viremia. Furthermore, the assembly of these reads generated only a 349 nt contig and 55% identity. The same was observed for the other families, which either had low coverage of reads, did not form contigs, or formed short sequences with low identity.

The *Iridoviridae* family has insects, fish, and amphibians as known hosts. The *Poxviridae* family infects birds, reptiles, mammals, and even humans [37]. For the *Iridoviridae* family, the contigs formed presented identities lower than 40%, and further analysis indicated their association with bat proteins through BlastX at NCBI. For the *Poxviridae* family, further analysis also showed identity with bat sequences. In addition, the standard number of reads for this family was very low, being less than 50 reads per pool.

*Phycodnaviridae*, *Iridoviridae*, *Mimiviridae*, and *Porxviridae* are viral families that comprise the phylum *Nucleocytoviricota*, known as large nucleocytoplasmic DNA viruses (NCLDV) [38,39]. A possibility for the identification of those families might be the fact that NCDLV can integrate eukaryotic genomes, and there are theories that propose the assimilation of parts of their genomes by protoeukaryotes [40]. Thus, the positivity found for these viruses may be related to parts of their genomes that also compose the genome of bats.

The families Nairoviridae, Nimaviridae, Rhabdoviridae, Iflaviridae, and Permutotetraviridae, although present in reads of bank matches, did not form contigs, while the families Polycipiviridae, Polydnaviridae, Picornaviridae, and Herpesviridae formed small contigs (>500 nt) that, in further analysis, did not confirm virus positivity.

The *Retroviridae* family was the most abundant in reads. The understanding of this viral family has grown a great deal due to the study of sequences that are integrated into the genome of chiropteran hosts due to the viral mechanism of insertion of its cDNA into the genetic material of the host, which occasionally ends up attaching itself there, giving rise to the so-called “endogenous retroviruses” (ERVs). The first associations between retroviruses and bats occurred precisely in cell lines of molossid bat lung (*Tadarida brasiliensis*) [41,42]. Most ERVs are “genetic fossils”, meaning they are inactive. However, some may still possess the ability to transcribe active elements or may reactivate in a way that is harmful to the host’s health [43]. ERVs might even possess genes commonly identified in exogenous retroviruses, such as gag, pol, pro, and env [44].

There were many contigs formed for the *Retroviridae* family, ranging from 190 to over 600 among the pools. Contigs that had a size above 900 bp and a 65% identity were evaluated using BlastX. Most of the sequences in this second analysis had a higher identity than endogenous retroviruses or sequences from other mammalian organisms.

### 4.2. Morbillivirus

The first appearance of the association of *Morbillivirus* with bats in the literature was a study published in 2012 [6]. Several genomic sequences were classified as “morbilli-related”, coming from several bat species, such as *Desmodus rotundus*, *Myotis myotis*, and *Coleura afra*, and several countries, such as Germany, Ghana, and Brazil. Two of the sequences found in the mentioned study came from spleen samples of bats of the species *Desmodus rotundus* that were collected in Brazil in 2008 and 2009. According to the phylogenetic tree assembled, these two sequences would be phylogenetically closer to *Phocine morbillivirus*.

In another work published in 2022 [45], the genomes of two Morbilliviruses were found and named *Myotis bat morbillivirus* and *Phyllostomus bat morbillivirus*. The samples used came from bats of the *Myotis riparius* and *Phyllostomus hastatus* species. The *Myotis* samples were collected in the Brazilian Amazon region, in the state of Amazonas in December 2013. Those of *Phyllostomus* were collected in the Atlantic Forest region in June 2013.

These findings indicate that the circulation of *Morbillivirus* species in bats from Brazilian forest regions, both Amazonian and Atlantic, has been occurring for almost a decade, so we can infer that viruses of this genus have found a favorable environment and opportunity to spread and maintain a cycle of infection, at least among bats. However, the way this circulation occurs and whether there is any correlation regarding the origin of circulating viruses remain to be determined.

Another fact to be noted is that, in these studies, the samples where the viral sequences were found were from rectal swabs, indicating that *Morbillivirus* in bats may be infecting both the respiratory and intestinal tracts. It is important to note that these systems may also serve as excretory pathways for the virus, so its transmission becomes more likely. These findings are consistent with those reported by Black (1991) [12], who points out that morbilliviruses cause respiratory and gastrointestinal infections. In addition, given the continuous contact of humans with a variety of bat species both in urban and rural areas, there is the possibility of transmission to humans in a possible spillover event.

A *Morbillivirus* infection can occur through the respiratory tract when the host inhales viral particles in suspension [12]. Given the habitat of bats, with the formation of colonies and constant exposure to an environment where feces and urine are present, infection through aerosols with viral particles and contaminated feces may be important routes of transmission for Morbillivirus amongst these animals. There are also precedents for vertical transmission of Morbillivirus species [46].

To better assess the possibility of human infection by bat morbillivirus, it is worth looking at the results presented by Lee and colleagues (2021) [11]. In the study, the ability of *Myotis bat morbillivirus* (MBM) to use cells expressing the CD150/SLAMF1 and NECTIN-4 receptors to replicate is evaluated. The former is expressed by immune cells, and the latter by epithelial cells. The results show that the virus has limited ability to use CD150 to infect human and dog cells, presenting a higher affinity for CD150 expressed by bats. As for NECTIN-4, the MBM was able to replicate in human lung cells that have a higher expression of this receptor.

Considering that the CD150 receptor shows greater variability between species and that NECTIN-4 is more conserved [47], it makes sense that Morbilliviruses coming from bats would be able to infect human epithelial cells that express NECTIN-4 but would not have the same success with cells expressing CD150 as this receptor would be the major one responsible for the specificity of *Morbillivirus* species for their hosts. It is fair to say that the occurrence of human Morbillivirus infections from bats seems to have little potential to cause great harm to those infected given the difficulty that the virus would have in replicating and spreading throughout the human body, especially if we look at the fact that, in *Measles morbillivirus* infection, cells expressing CD150 play a vital role in the systemic spread of infection. In addition, it is worth stating that the human-interferon-related defense process seems to be quite efficient in fighting a possible chiropters Morbillivirus infection, and that the eventual success of these viruses in human hosts would very much depend on improving their ability to resist or evade human host defenses [11].

The infection cycle of a *Morbillivirus* usually begins by entering cells that display CD150 (or SLAM-F1), such as activated lymphocytes, macrophages, and some types of dendritic cells. The virus then reaches secondary lymphoid organs, immune tissues, and then invades epithelial tissues via NECTIN-4 [48]. Taking this infection cycle into consideration, the finding of a *Morbillivirus* genomic sequence in bat lung may indicate that the virus managed to affect the entire infection pathway in the animal, even though it is not possible to determine if it can cause pathology, particularly due to the failure to obtain viral particles from the tissue used, and further studies are needed to elucidate this issue.

### 4.3. Dicistroviridae

Currently, the *Dicistroviridae* family is composed of the *Aparavirus*, *Cripavirus*, and *Triatovirus* genera [39]. This family presents a low degree of similarity among its species. Regarding its structural polyproteins, the aminoacidic similarity varies from 37% to 80% [49]. Moreover, some studies have already performed allocations of new viruses within this family with amino acid identity percentages lower than 40% [13,15]. Based on the nucleotide (89%, 83%, and 51%) and amino acid (94, 91%, and 42%) identity values of *Culex dicistrovirus* 2, *Rhopalosiphum padi*, and *Aphid lethal paralysis* viruses, the presence of common protein domains, and phylogenetic inference, it was possible to classify a contig within the genus *Cripavirus*, as well as other reference sequences classified in ICTV.

The detection of viruses known to infect insects and other arthropods in the feces of vertebrate animals can occur due to the process of insect oral entry into the studied animal, either by chance or on purpose due to food characteristics, which ends up reaching the intestine, and then, if the insect is carrying a dicistrovirus, it is detected in stool samples. When it comes to detection in tissues and blood, this justification seems more implausible. In those cases, we can discuss an interspecies barrier crossing occurrence, where the contact of these viruses with other hosts may have eventually led to an adaptation of them in these animals. However, in order to verify this possibility, studies involving viral isolation from vertebrate animal samples would be necessary. Even more complicated is to conclude whether the detection of these viruses is related to the presence of a disease process in the vertebrate host, especially when it comes to bats, animals that are known to carry viruses without having an associated clear pathogenic process. This would also require studies to prove the occurrence of pathogenicity in bat cells caused by dicistroviruses in cell culture. There has already been an attempt to culture a dicistrovirus from contaminated liver samples using vertebrate (Vero E6) and invertebrate (C6/36) cells, but there was no success [13].

The papers reviewed for the present work that detected dicistroviruses in chiropterans encompassed especially insectivorous bat species [13,16,17,18], but also fruit bats [19]. Frugivorous bats, although not as a primary goal, also end up ingesting insects through fruits due to the close relationship of these arthropods with this type of food [19]. Thus, it is valid to say that these groups of chiropterans have greater possibilities for a virus originated from insects to end up adapting to their organisms due to the constant interaction between them, which enhances the chances of a spillover event.

Another worthwhile point to mention is that bat species where dicistroviruses have been shown to be present involve the genera *Molossus* sp. and *Pipistrellus pipistrellus* [16,18]. These chiropterans are commonly cataloged near human habitations [50,51]. Thus, it is possible to imagine a scenario where the human population may end up coming into contact with viruses of the *Dicistroviridae* family, not only through insects but also through the feces of bats, for example.

Some studies have already reported the presence of this viral family in human serum samples. In these studies, the patients tested had febrile states. None of the research has ever determined whether the presence of the *Dicistroviridae* family viruses in these patients was involved in any way with their febrile condition or their ability to replicate in human cells [52,53].

## 5. Conclusions

The metagenomic analysis of lung samples from bats (*Malossus* sp.) captured in different states of the Brazilian Amazon region revealed the presence of several viral families. However, none of these families showed the presence of a virus with known significant medical importance in humans or animals, and there was no discernible correlation between the viruses of different samples, their location, or year of collection.

Nevertheless, this research showed a partial genome of a *Morbillivirus* that may represent a new species, identified by its similarity to the *Porcine morbillivirus* and *Phyllostomus bat morbillivirus* genomic sequences. Additionally, the presence of a genomic sequence of a possible new strain of *Culex dicistrovirus* 2 suggests a possible association with the feeding habits of these bats.

Overall, metagenomic surveillance studies are important tools for identifying and monitoring viruses in bat populations, with the aim of predicting outbreaks of viruses of medical importance.

## Figures and Tables

**Figure 1 microorganisms-12-00593-f001:**
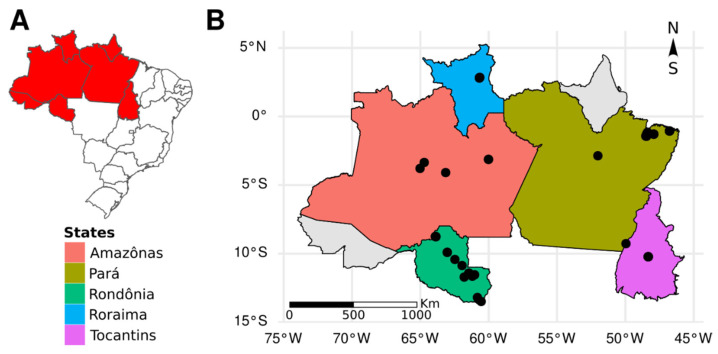
Identification map of the states and marking the collection sites of the bats used in the analyses. Legend: (**A**) in red is the north Brazilian region, which includes the states where the bats were collected; (**B**) states that had bat samples included in this work and the marking in black dots of the places in which the samples were collected. The maps were assembled using data from IBGE (2020) and with the programs ggplot2 and geobr.

**Figure 2 microorganisms-12-00593-f002:**
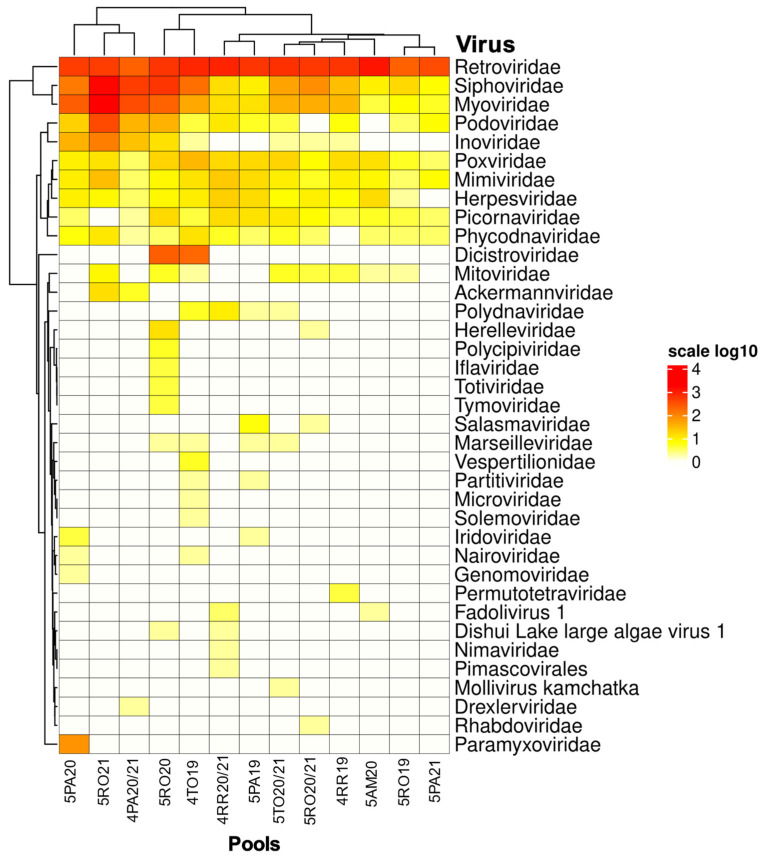
Heatmap for the reads of each pool related to their classification by viral family.

**Figure 3 microorganisms-12-00593-f003:**
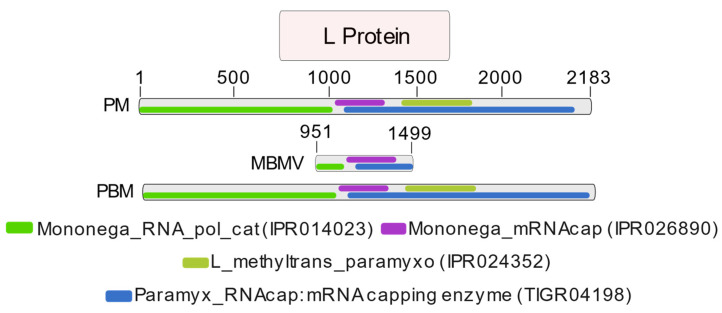
Identification of the MBMV protein domains, their comparison with domains in the same region of the *Porcine morbillivirus* genome, and positioning in the L protein ORF. Legend: PM = *Porcine morbillivirus*; MBMV = *Molossus bat morbillivirus*; PBM = *Phyllostomus bat morbillivirus*.

**Figure 4 microorganisms-12-00593-f004:**
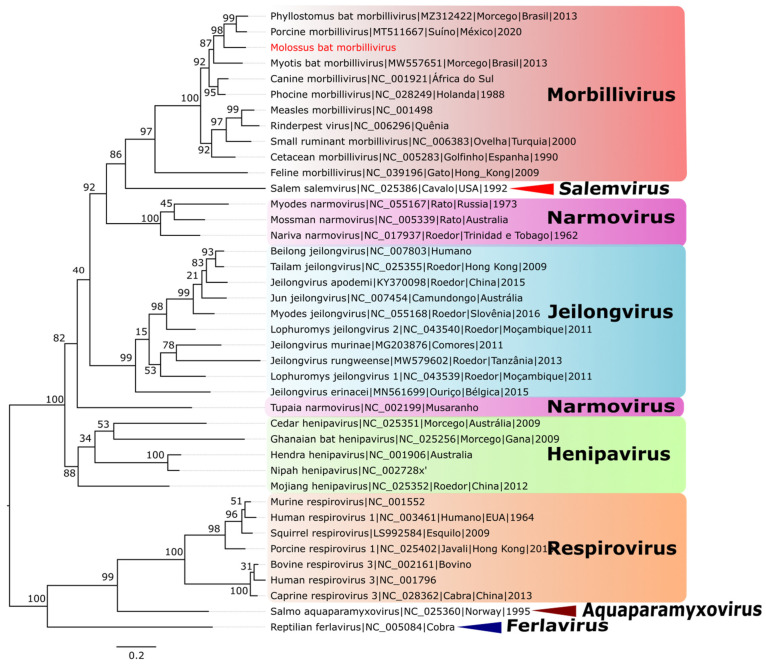
Phylogenetic tree of the *Paramyxoviridae* family. Legend: the Maximum Likelihood (ML) method was applied based on the amino acid sequences of a partial L protein obtained from metagenomic analysis. The LG+F+I+G4 model was used as the best amino acid substitution model. The sample identified in this study is highlighted in red. The different genera are highlighted by colors. The numbers at each main node in the tree correspond to bootstrap values in percentage (1000 replicates). The scale bar corresponds to the amino acid divergence per site between sequences.

**Figure 5 microorganisms-12-00593-f005:**
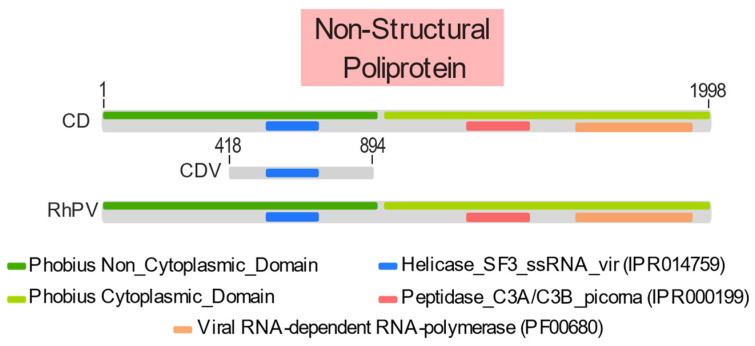
*Culex dicistrovirus* 2 5RO20/CDV protein domains, their comparison with domains in the same region of the *Culex dicistrovirus* 2, and *Rhopalosiphum padi virus* genomes and their positioning in the non-structural polyprotein ORF. Legend: CD = *Culex dicistrovirus* 2; CDV = *Culex dicistrovirus* 2 strain 5RO20; RhPV = *Rhopalosiphum padi virus*.

**Figure 6 microorganisms-12-00593-f006:**
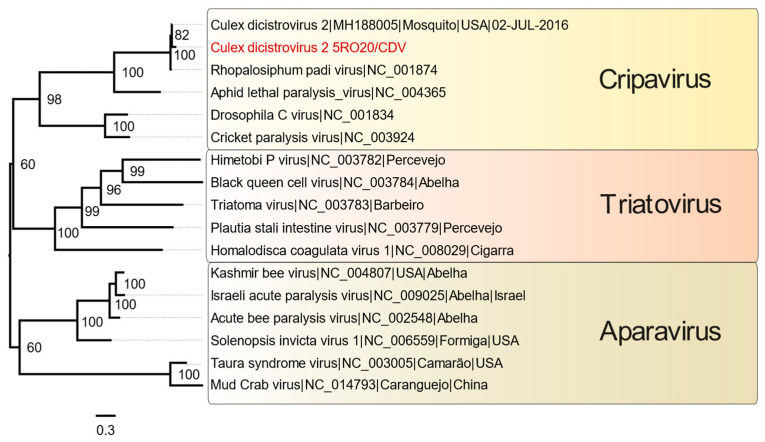
Phylogenetic tree of the *Dicistroviridae* family. Legend: the Maximum Likelihood (ML) method was applied based on the amino acid sequences of a partial non-structural polyprotein region containing 1454 nucleotides obtained from metagenomic analysis. The VT+F+R4 model was used as the best amino acid substitution model. The sample identified in this study is highlighted in red. The different genera are highlighted by colors. The numbers at each main node in the tree correspond to bootstrap values in percentage (1000 replicates). The scale bar corresponds to the amino acid divergence per site between sequences.

**Table 1 microorganisms-12-00593-t001:** Total reads generated by sequencing and remaining after each step in the data cleaning process.

Pool	^1^ Pool Identification	Raw Reads	After *Fast*p	After Host Genome Sequences Removal	After SortMeRNA
Pool 1	5AM20	29,907,996	27,369,014	8,950,902	8,788,973
Pool 2	5PA19	76,284,194	71,366,180	7,303,274	5,792,842
Pool 3	5PA20	80,318,832	70,481,544	11,806,566	10,503,106
Pool 4	5PA21	78,606,736	68,321,968	9,240,938	7,950,931
Pool 5	4PA20/21	77,126,060	67,638,454	8,610,316	7,034,313
Pool 6	5RO19	36,750,816	32,750,320	3,913,790	3,665,172
Pool 7	5RO20	36,711,948	33,125,972	8,523,712	7,023,625
Pool 8	5RO21	108,272,434	96,549,530	14,148,222	11,945,547
Pool 9	5RO20/21	22,753,752	20,550,060	5,807,698	5,697,788
Pool 10	4RR19	59,314,230	52,754,680	10,355,118	9,924,522
Pool 11	4RR20/21	85,614,884	79,378,240	11,628,984	10,669,511
Pool 12	4TO19	36,099,760	33,144,540	7,288,282	5,801,459
Pool 13	5TO20/21	103,219,948	91,761,254	11,082,910	10,401,188

^1^ Pool identification: the initial number consists of the number of bats, the letters correspond to the state sample origin, and the final numbers correspond to the year of collecting (e.g., 4PA20/21 to the pool of four chiropterans from Pará State, collected in the years 2020 and 2021).

**Table 2 microorganisms-12-00593-t002:** Total contigs generated and positive for viruses using the treated reads and the MEGAHIT and SPADES assemblers.

Pools	Contigs Megahit	Contigs Spades	Contigs Identified for Viruses
**5AM20**	58,144	465,165	2885
**5PA19**	45,984	294,280	2561
**5PA20**	59,724	354,512	2911
**5PA21**	38,614	221,168	1633
**4PA20/21**	32,277	190,489	4579
**5RO19**	20,548	152,333	1208
**5RO20**	63,143	443,249	4857
**5RO21**	66,122	341,695	8452
**5RO20/21**	57,360	437,210	2207
**4RR19**	72,500	461,112	2647
**4RR20/21**	111,111	663,004	4182
**4TO19**	44,285	381,686	3521
**5TO20/21**	79,031	450,534	3169
**Total**	748,843	4,856,437	44.812

## Data Availability

The data generated in this work are available at NCBI under the BioProject identification PRJNA956836. The SRA archives are identified by the accession identifications SAMN34231019 to SAMN34231031. *Molossus bat morbillivirus* nucleotide sequence is available at NCBI under the accession number OQ891082, and *Culex dicistrovirus* 2 strain 5RO20/CDV is available under the accession number OQ915114.

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
