# Peer review of "Identification of Viruses in Molossus Bats from the Brazilian Amazon: A Descriptive Metagenomic Analysis"

_microorganisms, 2024, doi:10.3390/microorganisms12030593_

Round 1
Reviewer 1 Report
Comments and Suggestions for Authors
This is an excellent study on bat-derived viruses, and metagenomic analysis has effectively detected a wide range of viral reservoirs in bats, making this manuscript of significant interest. However, there are some minor problems that need to be corrected, as commented below:
Comments:
1. Line16, 31, Line 34, and Line 63: Molossus, Molossidae, and Molossus sp.should be italicized.
2. Line 169, MV, The acronym 'MV' should be corrected to 'ML'.
3. Line 205-206, The name of the viral family should be in italics
4. Figure 4 and Figure 5. Clarifications are needed regarding the gene sequences' acquisition methods—whether they were obtained through re-PCR and re-sequencing or directly from metagenomic results. Additionally, specify if the gene sequences used in constructing these phylogenetic trees are complete or partial, and whether they match the length of the reference sequences. It is advisable to include these details in the legends of the figures.
5. Because this study involves the collection of animal samples, whether the study has received animal welfare and ethics-related reviews or approval from relevant organizations?
Author Response
Thank you for taking the time to review this article and for the comments to improve our work.
Comments 1, 2 and 3: The requested corrections were made and are marked in yellow at the document of the revised manuscript.
Comment 4: The requested correction was made and are marked in yellow at the document of the revised manuscript.
The sequences were obtained directly from metagenomic results.
The sequences obtained are only partial, not covering the entire length of the gene.
Comment 5: Ethical review and approval were waived for this study due to the bats used in this research were sent dead to Evandro Chagas Institute, as part of the rabies control program carried out by Brazilian health agencies. Therefore, there was no need to request authorization from the institution's ethics committee, as no animal was killed in order to collect the lung samples.
This same paragraph clarifying the ethical permissions for the study was added to the last page of the article.
I hope that the answers and corrections I've made have cleared up any doubts or problems identified in the work, and i thank you once more for the time spent in correction of this work and the contributions made to its improval.
Reviewer 2 Report
Comments and Suggestions for Authors
The authors present the results of a comprehensive descriptive study of viruses present in the lungs of 58 Molossus bats that were collected in five different regions of the Brazilian Amazon. Their sophisticated methodology is described in detail and the tentative identification of multiple virus genera and species is state of the art.
My comments on the manuscript are minor and any modifications in the text are up to the authors.
I think a more comprehensive and appropriate title might be, “Identification of viruses in Molossus bats from the Brazilian Amazon: a descriptive metagenomic analysis”.
In my opinion the authors’ conclusions can be toned down to bee more understated. They may want to modify statements for which there is scant evidence. Given the proximity of bats to human populations continued analysis of their viral repertoire is certainly necessary. However, I think it is too speculative to state, as written in the Abstract that, “we highlight the importance of these studies….” Their study is descriptive and clinical relevance for human diseases remains undetermined and is largely unstated.
Author Response
Thank you for the time spent in correction of this work and the contributions made to its improval
We agree with the suggestions made. The article title was altered to the title suggested and the abstract phrase pointed out related to the conclusions was excluded from the article.
Reviewer 3 Report
Comments and Suggestions for Authors
- The article overall is very well written and presented.
- I suggest that you review the writing of the viral nomenclature by placing the family names WITHOUT italics and using italics only for viral genus and species. I highlighted several examples in comments in the text.
- Considering that there was a detection of another Dicistrovirus in bats in Brazil (ref.13), why did you not include this virus in the phylogenetic tree with the other Dicistroviruses? Have you assessed whether they have any similarities?
- Although I cannot see, based on the discussions in this article, the importance of these identified viruses for human and animal health, I think that metagenomics is really a tool of great importance and bats, hose bearers, deserve great attention as they are potential triggers of spillover events that could have disastrous consequences in the future.

Author Response
Thank you for the time spent in correction of this work and the contributions made to its improval.
- Thank you for the comment about the writing of the viral nomeclature. The names of the viral families are written in italics because we used the ICTV guidelines as a basis, which indicate that the names of the families should be written in italics.
- The Dicistrovirus sequence related to the article of the reference 13 was not included at the phylogenetic tree because attempts to include this sequence caused an alteration of the genuses organization as seen in ICTV. In addition, even in the phylogenetic tree containing this sequence, the genomic sequence indentified on our work still form the same clade in the same genus, with the sequence of the reference 13 being agroupped at other genus. So we chose to use the phylogenetic tree without the mentioned sequence. Please see the attachment where we uploded the mentioned tree.
I hope that the answers may have cleared up any doubts or problems identified in the work.
